# Family Social Support and Children's Mental Health Resilience during COVID-19—Case of Morocco

**Mohamed Kadiri**

Faculty of Letters and Human Sciences, University Ibn Zohr, Agadir 80000, Morocco; m.kadiri@uiz.ac.ma

**Abstract:** This research aims to investigate the impact of the family on the social support for children with mental health disorders in a vulnerable environment such as the COVID-19 pandemic and the ability of parents to provide consistent care giving using their social support. The main question of this research paper is: what is the impact of the family in the situation of COVID-19 on the diversity of the processes actors provided in creating social adaptability for children with mental health? The previous question was treated via the following hypotheses: H1: the more the whole family is committed to the social support of the child with a mental health disorder, the less independent the achievement of resilience on an individual basis; H2: the more the child with a mental illness belongs to the family, the greater the family's involvement in social support, even for low-income families. This quantitative field study was conducted in southern Morocco by contacting a sample of respondents (86 respondents) from health institutions, civil society, and the children's parents and relatives. The study reflected two main findings: that the majority of children were with their families during the pandemic; and the lack of independence of parents and children in the adjustment process.

**Keywords:** social support; family resilience; individual resilience; child mental health; COVID-19

## 1. Introduction

The role of the parents or guardians is crucial in caring for children and seeing to it that their requirements for food, clothes, education, shelter, and protection are addressed, particularly in times of crisis. In addition to these responsibilities, if the children suffer chronic psychological disorders or mental illness, this requires additional work and duties. The presence of the family thus becomes vital in ensuring that these children receive social assistance [1], particularly since researchers have confirmed that mothers of mentally ill children have a two to three times higher risk of developing depression than mothers of healthy children [2].

Many forms of solidarity, such as those between families, extended families, associations, and other institutions, appeared during the COVID-19 disaster. Disasters are generally defined as events that create significant disturbance to the daily lives of a large number of people. In addition to discussing various experiences about the collective trauma they produce, the research on the effects of disasters on children covered the operations of interactions linked to social support intended to assist the resilience and development of these children [3].

The social position of the children, their age, and the overall environment characterized by the risks of COVID-19 make these social strata, particularly those with mental health disorder, more vulnerable [4].Given that there are circumstances that can exacerbate this vulnerability with the whole family, such as poverty, the growth of its members, divorce, and the parents' low educational status, all of these situations are added to the confinement context concerning the needs of COVID-19; these conditions can negatively affect children's abilities and well-being by making it difficult for parents or other caregivers to care for their health [5]. They may also make it difficult to obtain daily supplies and necessities, which may eventually result in abuse, neglect, and violence against children with mental illness [6,7].

Social support is significantly associated with resilience through psychosocial mechanisms for having the ability to cope with stress. Moreover, social support is one of the crucial key factors of individual resilience during the COVID-19 [8,9]. Indeed, studies demonstrate that social support from family and other loved ones was associated with greater resilience [10]. The positive environment during the COVID-19 provides a factor of satisfaction at the level of psychosocial needs such as affiliation with others, affection, and social support from family, friends, and peers [11]. Furthermore, social support services are correlated with improved well-being and positive family functioning and support the social integration of people with dementia, specifically at the level of reduced illness [12]. The present study attempts to cover the scientific gap that occurred in the study of the mental health of children suffering from mental illnesses before COVID-19 because, in light of the abundance of studies that dealt with the mental health of family members, especially children, during the pandemic, it is matched by a scarcity, as far as we know, in studies that tracked the mental health of children with mental illness before the pandemic. The current topic finds its basis in the relationship between the concepts of social support and resilience on two levels. The first is the level of scientific specialization represented by the psychosocial field, which promotes the understanding of social realities through the interaction between the individual and the group. The second level is represented by the situation or "the life event", which is the COVID-19 period, in which it is not possible to understand what is happening in it except by understanding what happened before it, particularly at the level of social support, which significantly affects resilience, particularly among children with mental illnesses. Although the concept of social support did not pose any difficulty, social resilience is required to address resilience (as a concept) in its various psychological manifestations (individual resilience, child resilience) and social expression (family resilience).

## 1.1. Social Support

Although many authors have used terminology that suggests they are discussing the idea of social support, after the term was adopted in the 1970s, questions about its content, the social context surrounding it, the spread of theoretical definitions, and discussions that refer to it emerged [13,14].

It was challenging to conceptualize, illustrate, or measure the diverse visions of social support. Despite the extensive studies that have been conducted on it, there has yet to be a consensus among theorists and researchers as to what it means in theory and practice. As a result, social support researchers ignored the complexity of this concept, used an explicit measurement method, and assumed that practically everything that necessitates social interaction qualifies as social support [14,15]. One study collects definitions of social support. It condensed them into two categories based on characteristics. The first category describes the dimension of interpersonal relationships, including structure, size, reciprocity, accessibility, make-up, and intention. The second category defines the dimension of social ties, including access to, use of, homogeneity, motivations, and expectations. This study ends with one element, the social relationship that ties the two dimensions together. It is characterized by structure, type, and strength on the one hand, and reciprocity, accessibility, and reliability on the other [15].

In the definition provided by Shifeng Li and Qingying Xu, "social support is typically conceptualized as the emotional, informational, and instrumental assistance received from social network" [16] (p. 2).In addition, Dunst, Trivette, and Cross confirm that "social support is a multidimensional construct that includes physical and instrumental assistance, attitude transmission, resource and information sharing and emotional and psychological support" [17] (p. 98).

To better understand the concept of social support, researchers suggest that it includes two fundamental components: perceived social support and the received social support [18]. Each of these components corresponds to a unique perspective on social support. Received social support correlates with the view that social support results from each individual's special characteristics. It is established through early interactions between parents and

children. Moreover, it is characterized by temporal stability. This approach is founded on the fundamental idea that social support is directly related to intrapersonal processes and internal relational schemas or interactions that take place before the person in question asks for assistance: the interrogatory process [7,19]. By contrast, the second approach, which refers to received social support, considers social support due exclusively to transactions that concern the social environment or the resources the individual can access. This approach is based on the assumption that social support is interpersonal or interactions that take place following a request of support: the postrogatory process [7,19].

In measurement, the perceived social support originates in early family transactions that include caring, affection, and positive involvement which set the basis for supportive relational schemas. Furthermore, researchers have found a relationship between perceived social support and infant attachment, family environment, and social skills. Conversely, it also correlated to feelings of control, self-efficacy, and self-esteem. As for received social support, it is less dependent on the previous stage, which is related to the parents, but instead has to do with situational factors. This situation is characterized by the fact that the person concerned is under stress and needs help to cope [7,18,19].

Ambikile and Outwater measured received social support through three needs that parents and guardians require in their relationship with children with mental disorders: psychological needs, which include disturbing thoughts, emotional disturbance, unavoidable situations, and communication problems; social needs, which include social service, stigma, caring responsibilities, public awareness, social support, and social life; and economic necessities, which refer to poverty, interference with various income-generating activities, and extra expenditure due to the illness [1].

Schaefer and his colleagues distinguished between three functions of support: Emotional support, which includes intimacy, affection, consolation, and the ability to trust and rely on others—all of this contributes to the person being valued, cared for, and a member of the group rather than a stranger; tangible support, which includes direct assistance or services such as caring for people in need or doing their routine work, as well as loans, financial assistance, or goods; and informational support, which refers to delivering information and suggestions to help the person solve the problem and give feedback on his performance [20]. Note that researchers have proved that emotional support is more critical than informational support and tangible support, as it contributes significantly to maintaining good health and a sense of well-being [19,20].

Research on adults reveals that among the most effective indicators of long-term mental and physical health after a traumatic experience are the availability of dependable social support and community-based resources (such as employment chances). However, there as been limited empirical study on the impact of family coping mechanisms on adolescents exposed to significant disasters regarding posttraumatic distress. One possibility is that parents may be forced to choose between different coping behaviors due to their limited time and energy. For example, they may have to choose between offering their children higher levels of intimate social support (such as talking about their feelings) and seeking agent-based, logistical support (such as professional counseling, job training, housing assistance, and the like) [21].

### 1.2. Resilience

Invulnerability was the first concept in this area to emerge. It grew in popularity in the mental health field in the 1970s. This idea is seen as a logical progression from the psychological consequences of what was once called Stressful Life Events and Circumstances (SLE-Cs). The invulnerability was applied to a small sample that defies the odds in a group of well-adjusted and healthy black children raised in the community that they could tolerate despite the extreme stress brought on by deprivation, poverty, and prejudice. Although this concept's first formulation is just the beginning of its examination, the idea has proven helpful as a predictive tool for highlighting similarities among various datasets [22,23].

The concept of resilience has replaced the idea of invulnerability because it does not help to expand the research; it gives the impression that the person concerned is safe and would not be exposed to risks and that it is the personal characteristics in him that protect him. It also gives the impression that he covers all risks by dealing with vulnerability as a constant characteristic; it fails to account for the dynamics of the social environment and the different personalities [24,25].

Scientists and clinicians have sought to learn more about how adversity impedes human adaptation and how to reduce risks such as those left over from World War II [24].

Understanding the factors that encourage healthy coping or reduce the impact of risk or adversity was critical to the Child and Family Study scientists, who confirmed that some individuals or families appeared more vulnerable to problems. They also note that some people have better protection or recovery than those who have experienced trauma or crisis [26].

Definitions of resilience have evolved over decades of research on psychosocial resilience, reflecting a general theoretical movement toward developmental systems models in conceptualizing and studying human adaptability and development. It spans a variety of interactions, starting at the molecular level and moving on to social and ecological systems [27]. According to Virginia O'Leary, the first definition of resilience originated in the physical sciences—it refers to how solid materials react to the direct treatments they receive—but the second definition is psychological and is more widely accepted by social scientists [28].

Resilience has not been used consistently in the scientific literature despite its intuitive appeal, leading to inconsistent definitions among researchers: definitions have included academic achievement and social competence [23], overcoming difficulty and surpassing disadvantage [29], the positive adaptation in the difficult conditions, achievement of social expectations, and the capacity to change behavior in response to various demands. In the end, Werner and R.S. Smith (1982) focus on the ability to handle stress [28].

Indeed, this is what Masten noted when she confirmed that even for research on individual resilience in young people, there are inconsistencies in the body of research on resilience due to the complexity of the processes involved [27].

Resilience was conceptualized by Rutter (1987) as a protective mechanism that fosters effective adaptation in the face of psychological or environmental stressors. The four roles that resilience serves are to (a) lessen risk impact; (b) stop adverse domino effects; (c) build and preserve self-esteem and efficacy; and (d) improve opportunities [29].

Garmezy mentioned that "Resilience means the skills, abilities, knowledge, and insight that accumulate over time as people struggle to surmount adversity and meet challenges. It is an ongoing and developing fund of energy and skill that can be used in current struggles" [30] (p. 298).

*1.3. Family Resilience*

Few studies thought of the family as a potential source of resilience, even though individual resilience has increasingly come to be understood in terms of the interaction between nature and nurture [31]. Indeed, it has been emphasized by some researchers that only since the 1970s studies and research have started to examine families' support for each other, their capacities, and roles for coping and adaptation [32].

The literature assessment reveals at least two analytical frameworks that combine family and resilience [33]. Sometimes, the family is considered a risk factor for resilience because it is the source of stress; simultaneously, it is seen as a protective factor since it is the individual's source of resilience [34].

The idea of the family as a source of resilience is related to the concept of the family environment, which is made up of two elements: total family stress, which is defined as the number of life events the family experiences in a given period (such as unemployment, illness, and financial crises); and total family functioning as a positive factor (minimal conflict in the home during infancy; absence of divorce during adolescence) [35], which refers to the family as a whole and not just one person or a small system. Moreover, the fathers may participate in achieving resilience through their contribution to the childcare in the family [36,37].

Finally, in some cases, the family is considered a risk factor for children, as it can participate in specific behaviors linked to the emergence of mental illness and crime in children such as via severe marital conflict, maternal mental illness, crowded housing, and limited parenting skills [38,39].

### 1.4. Individual Resilience

Individual resilience depends on the qualities of people who have overcome adversity and on the consideration that some have an advanced level of resilience. Therefore, resilience is typically seen as the personality traits and coping mechanisms that allow a child or adult to overcome traumatic life experiences. Accordingly, resilience is typically seen as inborn, as if resilient people were raised as such from birth and always had it—i.e., biological hardiness—or they acquired it on their initiative and through good luck. The term "vulnerable child" has been used, contributing to the perception that fragile survivors of abusive home contexts are resistant to stress due to their internal resilience or personal armor [31,40].

Studies have revealed that characteristics such as a cheerful and composed temperament and high intelligence are advantageous but not necessary for developing resilience. Most importantly, these characteristics lead to increased self-esteem, defined by a realistic feeling of hope and personal control [31,35].

According to Kobasa and colleagues, people with hardy personalities share three traits in common: the conviction that they have some degree of control or influence over the circumstances of their lives; the capacity to feel intensely invested in or committed to their daily activities; and the expectation that change presented an exciting opportunity for personal growth [10].

Polk (1997) looked into four methods to empirically assess individual resilience. The first, the dispositional pattern, refers to the biological and ego-related psychosocial attributes that contribute to the manifestation of resilience. The second, the relational pattern, concerns the person's social obligations and interactions with others. These roles and connections involve close, personal relationships and more societal connections. The third pattern is the situational pattern; it deals with the elements that concern the connection between a person and a difficult circumstance. This can include the person's capacity for problem-solving, evaluation of conditions and possible answers, and action in the face of challenge The final type of resilience, the philosophical pattern, is a person's perspective on the world or way of living. This can contain various resilience-promoting ideas, such as the conviction that there is positive significance to be discovered in every experience, the importance of self-improvement, and the belief that life has meaning [32].

### 1.5. Children Resilience

Breda has noted that the initial research on individual resilience concentrated on children's capacity [7]. Taking into account that epidemiological studies are concerned with health and disease in human populations, the contemporary notion of risk in biological and behavioral studies in particular focused on the factors at the person's disposal for mental and physical health. Indeed, the research on life stress and risk established the foundation for the later theoretical framework for resilience [26].

Biological risk factors have been the main reason behind studies in the 1930s about children exposed during prenatal and postnatal periods, including nutritional deprivation, low birth weight, infections, and schizophrenia [41,42]. Additionally, environmental risk factors, including stress studies, were a focus of research in the 1950s to examine how children and adolescents develop in response to naturally occurring stressors [28].

Resilience was emphasized in the 1970s when academics redirected their attention from the negative effects of risk to beneficial outcomes such as adaptation, protection, competence, and invulnerability. The concept of "resilience" was first used by Gramezy and Nuechterlein (1972) to describe a small group of competent African American children living in the ghetto. These children could cope despite facing severe social and environmental difficulties such as poverty, prejudice, and unfavorable living conditions [28].

Resilient children were defined as children who manage to overcome these challenges under such circumstances. "Resiliency in children is the capacity of those of who are exposed to identifiable risk factors to overcome those risks and avoid negative outcomes such as delinquency and behavioral problems, psychological maladjustment, academic difficulties, and physical complications" [43] (p. 368).

On the Hawaiian island of Kauai, 700 native children born in 1955 were the subjects of longitudinal research by Warner and R.S. Smith in 1982. This study expanded on earlier investigations into infant development. The 10% of people the researchers classified as resilient shared four key traits: the first is an active approach to problem-solving; the second is a propensity to see their experiences as beneficial, even when these experiences result in pain or suffering; the third component is the capacity, beginning in infancy, to attract the favorable attention of others; the fourth key is a firm reliance on faith to maintain an optimistic outlook on a meaningful life [44].

The studies cited in this article focus on social support or individual or family resilience and its relationship with risk. Only some studies, to our knowledge, deal with all its components explicitly concerning COVID-19.

The main objective of this research is to argue that there is a close relationship between the individual, especially individual resilience, and the institution of the family as a whole that promotes resilience and adjustment for the mental health children. As a social institution, we expect the family to be both the means and the goal. It is considered a means because the child cannot achieve his independence without it. It is the goal because it is related to children with mental disabilities. Suppose they can integrate into the family. In that case, they could incorporate it into the overall social environment, therefore proving that the family is the primary source of resilience, whether it is individual or social, even in line with the proposal to return to it as a central institution in the adjustment of these children, regardless of its financial situation.

In this context, we ask the following main question: what is the impact of the family in the situation of COVID-19 on the diversity of the processes actors provided in creating social adaptability for children with mental health? We try to answer this question based on the following two hypotheses:

**H1:** *The more the whole family is committed to the social support of the child with a mental health disorder, the less independent the achievement of resilience on an individual basis.*

**H2:** *The more the child with a mental illness belongs to the family, the greater the family's involvement in social support, even for low-income families.*

## 2. Materials and Methods

Participants: The field research was conducted through a sample of respondents (86 respondents), those involved in caring for children with mental health, such as associations, philanthropists, extended family members, everyone who is familiar with the child or has a close connection to him, and the child's parents. After their approval, the form was immediately filled out, which included a set of variables consistent with the main problem adopted in this research. The respondents were selected randomly due to the nature of the small sample, which we found very difficult to assemble without the help of the associations and the employees in charge of carrying out social services for the benefit of this group. After we explained all the operational details to the association actors and respondents, the associations assured the respondents that the research was related to academic research which did not aim to study personal cases. Moreover, the nature of the questions included in the questionnaire did not reference what could identify the person concerned, his name, his family affiliation, or others which could harm the reputation of the respondents or their families.

The great city of Agadir and its surrounding urban and semi-urban areas, as well as the rural communities surrounding the city, were used as the basis for this study's sample of respondents. Because this region is situated in the south of the Kingdom of Morocco,

one of its essential characteristics that set it apart from the major economic cities in the country's north and center is its distance.

The proportions of the sample were as follows: mothers of children who suffer from mental illness 23 (27%); fathers of these children 9 (11%); their family members 12 (14%); civil servants and social workers in associations 25 (29%); and anyone who knows the child or has a close relationship with him 17 (20%).

The first time we encountered some difficulties in finding our sample, which occur because of the scarcity of families with a child with a mental health disorder. However, we raised this constraint by involving the associations in direct or indirect relations with children.

Outcome variables: We constructed this questionnaire (28 questions) and partitioned it following the theoretical components of our research. Indeed, we have constructed it in the form of five axes: the first is for the respondent description, which contains four questions (age, sex, and level of education); the second is for the child definition, consisting of three questions (age, sex, and does he have a family); the third is related to the socioeconomic conditions of the family, and it also consists of three questions (the economic conditions of the child's family, the place of residence of the child's family, and the presence of the child within the family during the period of COVID-19); the fourth part of the questionnaire focuses on the essential needs of the child, including economic coverage, nutrition, hygiene, and medical care, and it contains ten questions; the fifth and last part tried to evoke the relationship between the child, his family, and the conditions in the period of COVID-19 through eight questions.

**Analyses:** The procedure of the analysis that we have adopted is based on arguing the hypotheses from a set of tests that aim to verify the data, especially those that are statistically significant. For this reason, we used the SPSS software (version 22). We analyzed the data in two stages. The first concerned the descriptive analysis, which aims to explore most frequencies. We used the chi-square test in the second stage to verify the correlation between the variables studied following the significance point at 0.05 or 5%. However, for more precision, we proceeded to the post hoc analysis of the results since most of the nominal variables contained three answer indicators. This condition obliges us to check the significance of each variable within each significant effect of the chi-square. Therefore, we use the adjusted Z score to obtain the *p*-value of each variable. By dividing the global plus value from 0.05 to 9 (9 analyses emanating from the crossing of the two variables with three response indicators for each), we would have the frequency of 0.0056 as a new threshold for rejecting the null hypothesis.

In the tables presented in this article, we have included significant results at the level of the overall chi-square test. We have named it ($\chi^2$) and its *p*-value is displayed in the far right of the table. On the other hand, we have detailed the results in the table after calculating the significance of the nine response modalities within each variable. For this, we have named it (adjusted) and adopted the significance of the variables at this level.

### 3. Results

*3.1. Statitics Results*

3.1.1. Descriptive Statistics

It should be noted that the majority of children who suffer from mental illnesses are male (69%)—almost double the number of girls (31%)—and that a large proportion of them are aged between 3 and 13 years (55%). It is detected that most of the descriptive results confirmed the effort of all the families toward the children who suffer from mental health disorders. Indeed, the proportion of the children who have a family is 84%, and 90% of the sick children were with their families during the period of COVID-19. Moreover, a considerable proportion of these children were covered by their families. In addition, the father is the one who bears the financial burdens, and the mother carries the activities related to nutrition (69%), hygiene (88%), and healthcare (61%).

### 3.1.2. The Importance of Family Support

Checking the inferential results (Table 1), we notice the confirmation of the descriptive results since the important first significance is recorded with the variable that focuses on the sick child's belonging to all the family(father and mother)with a frequency of 60 answers (98%) ($\chi^2$ = 36.248, *df* = 4, *p* ≤ 0.0000), against a child who belongs to a family consisting only of his mother. In the second result, it is clear that a large proportion (72%) of sick children belong to a middle-class family ($\chi^2$ = 22.208, *df* = 4, *p* = 0.0002), followed by the proportion of children in low-income families (20%), with a high overall significance of the chi-square, and also at the level of the significance of the adjusted Z-score ($\chi^2$ = 22.208, *df* = 4, *p* = 0.0001).

**Table 1.** Frequencies and chi-square for the father support in the period of the COVID-19 pandemic by the variable of financial support = the father (*n* = 61).

| Variable | Family (Father and Mother) | | | Mother | | | Member of Family | | | $\chi^2$ | *p*-Value |
|---|---|---|---|---|---|---|---|---|---|---|---|
| | *n* | % | (Adjusted) | *n* | % | (Adjusted) | *n* | % | (Adjusted) | | |
| Child have a family | 60 | 98 | 0.0000 * | 1 | 2 | 0.0000 * | -- | -- | 0.0003 * | 36.248 | *p* ≤ 0.0000 |
| | Poor family | | | Middle-class family | | | A well-off family | | | | |
| Socioeconomic status of the family | 12 | 20 | 0.0001 * | 44 | 72 | 0.0002 * | 5 | 8 | 0.9759 | 22.208 | *p* = 0.0001 |
| | In the association | | | In a member of the family | | | In his family | | | | |
| Child residence in the period of COVID-19 | 2 | 3 | 0.8686 | -- | -- | 0.0001 * | 59 | 97 | 0.0007 * | 18.725 | *p* = 0.0008 |
| | Philanthropists | | | His family | | | The associations | | | | |
| Source of material expenses of the child | 1 | 2 | 0.0006 * | 60 | 98 | 0.0000 * | -- | -- | 0.0254 | 44.448 | *p* ≤ 0.0000 |
| Feed the child | 1 | 2 | 0.5096 | 60 | 98 | 0.1444 | -- | -- | 0.1161 | 10.928 | *p* = 0.0273 |
| Child care | -- | -- | 0.0014 * | 57 | 93 | 0.0219 | 4 | 7 | 0.8116 | 22.162 | *p* ≤ 0.0000 |

* *p*_value of the adjustment Z score = 0.0056.

The importance of the family during the period of COVID-19 is reflected by the result, which confirmed that a large majority of children were within their families (97%) during this period, with a very high significance($\chi^2$ = 18.725, *df* = 4, *p* = 0.0007).

In addition, the results reflect the significance of the variable concerning the sources of the child's financial expenses, which is the family at a rate of 98% ($\chi^2$ = 44.448, *df* = 4, *p* ≤ 0.0000). On the other hand, Table 2 reflected a limited number of children who were cared for by family members (8%), especially at the level of hygiene service (3%).The low results (8%) were also reflected in the denial of membership in the child's adjustment by the father's support alone ($\chi^2$ = 11.118, *df* = 4, *p* = 0.0015).

**Table 2.** Frequencies and chi-square for the father support in the period of the COVID-19 by the variable of financial support = the father (*n* = 61).

| Variable | At the Association | | | With His Family | | | With a Member of the Family | | | $\chi^2$ | *p*-Value |
|---|---|---|---|---|---|---|---|---|---|---|---|
| | *n* | % | (*Adjusted*) | *n* | % | (*Adjusted*) | *n* | % | (*Adjusted*) | | |
| The family prefers that the child stays during the period of COVID-19: | 4 | 7 | 0.6454 | 57 | 93 | 0.0069 | -- | -- | 0.0001 * | 17.942 | *p* = 0.0012 |
| | Member of the family | | | Father | | | Mother | | | | |
| Child care by: | 5 | 8 | 0.0051 * | 19 | 31 | 0.0233 | 37 | 61 | 0.9250 | 29.377 | *p* ≤ 0.0000 |
| Feed the child | 2 | 3 | 0.0100 | 17 | 28 | 0.1137 | 42 | 69 | 0.9384 | 18.567 | *p* ≤ 0.0009 |
| Child hygiene | 2 | 3 | 0.0027 * | 2 | 3 | 0.3597 | 57 | 94 | 0.0219 | 23.002 | *p* = 0.0001 |

**Table 2.** *Cont.*

| Variable | At the Association | | | With His Family | | | With a Member of the Family | | | $\chi^2$ | *p*-Value |
|---|---|---|---|---|---|---|---|---|---|---|---|
| | *n* | % | (*Adjusted*) | *n* | % | (*Adjusted*) | *n* | % | (*Adjusted*) | | |
| | Yes | | | No | | | Undecided | | | | |
| Child adjustment by the father support | 38 | 62 | 0.0261 | 5 | 8 | 0.0015 * | 18 | 30 | 0.8888 | 11.118 | *p* = 0.0252 |
| Child adjustment by the mother support | 24 | 39 | 0.0806 | 15 | 25 | 0.6473 | 22 | 36 | 0.1449 | 13.044 | *p* = 0.0110 |

\* *p*_value of the adjustment Z score = 0.0056.

In Table 3, significant results confirm that there is no independence in support for the father (No, 12%; Undecided, 76%)and of the mental health child($\chi^2$ = 56.535, *df* =4, *p* ≤ 0.0000). In addition, the results confirmed the dependence of the child (21%), and its limited ability to have an individual resilience ($\chi^2$ = 15.855, *df* = 4, *p* = 0.0021).

**Table 3.** Chi-square for the child adjustment by the mother's support in the period of the COVID-19 pandemic = yes (*n* = 58).

| Variable | In the Association | | | In a Member of the Family | | | In His Family | | | $\chi^2$ | *p*-Value |
|---|---|---|---|---|---|---|---|---|---|---|---|
| | *n* | % | (Adjusted) | *n* | % | (Adjusted) | *n* | % | (Adjusted) | | |
| Child residence in the period of COVID-19 | 1 | 2 | 0.1994 | 1 | 2 | 0.0059 | 56 | 96 | 0.0022 * | 25.552 | *p* ≤ 0.0000 |
| | Member of the family | | | Father | | | Mother | | | | |
| Feed the child | 1 | 2 | 0.0017 * | 16 | 27 | 0.1713 | 41 | 71 | 0.5488 | 21.400 | *p* = 0.0002 |
| Child hygiene | 2 | 3 | 0.0071 | 2 | 3 | 0.3201 | 54 | 94 | 0.0488 | 16.327 | *p* = 0.0026 |
| | Philanthropists | | | His family | | | The associations | | | | |
| Child care | 2 | 3 | 0.0645 | 55 | 95 | 0.0072 | 1 | 2 | 0.0636 | 11.055 | *p* = 0.0259 |
| | Member of the family | | | Father | | | Mother | | | | |
| Child care by: | 3 | 5 | 0.0002 * | 17 | 29 | 0.1286 | 38 | 66 | 0.1679 | 14.942 | *p* = 0.0048 |
| | Yes | | | No | | | Undecided | | | | |
| Child adjustment by father support | 7 | 12 | 0.1280 | 7 | 12 | 0.0000 * | 44 | 76 | 0.0000 * | 56.535 | *p* ≤ 0.0000 |
| Child adjustment by the family members | 16 | 28 | 0.3251 | 4 | 7 | 0.0000 * | 38 | 65 | 0.0001 * | 41.027 | *p* ≤ 0.0000 |
| Child adjustment by the philanthropists | 42 | 72 | 0.0410 | 6 | 10 | 0.0016 * | 10 | 17 | 0.4285 | 16.890 | *p* = 0.0020 |
| Child adjustment by the association | 33 | 57 | 0.0314 | 6 | 10 | 0.0126 | 19 | 33 | 0.7859 | 17.207 | *p* = 0.0017 |
| Child adjustment by the child himself (individual resilience) | 15 | 26 | 0.4103 | 12 | 21 | 0.0021* | 31 | 53 | 0.0299 | 15.855 | *p* = 0.0032 |

\* *p*_value of the adjustment Z score = 0.0056.

## 4. Discussion

The main objective of this research is to verify the factors likely to participate in the adaptability of children with mental illness during the pandemic period. This study should also give an idea of the tasks involved in this adjustment, taking into consideration financial cover, hygiene, food, and health care, on the one hand; and on the other hand, who is in charge of his tasks, whether the father, the mother, a family member, or the association.

This research has built its problem from an initial question that asks about the impact of the family in the situation of COVID-19 on the diversity of the processes provided by actors in creating social adaptability for children with mental health. We have tried to answer this question via two hypotheses. The first is that the greater the whole family's commitment to the social support of the child with a mental health disorder, the less independent the achievement of resilience on an individual basis. The second is that the more the child with a mental illness belongs to the family, the greater the family's involvement in social support, even for low-income families.

The first significant result is the one that confirmed that individual family members could not achieve resilience, and this was very clear, particularly for the father and the child concerned, which means that the importance of the institution of the family is not limited to the realization of the daily tasks carried out by its members, including the tasks related to the existence of a sick child. Still, the results go beyond this towards the importance of the presence of each member of the family.

This result confirmed the first hypothesis of this research: family resilience is the source of other resilience, specifically individual resilience. Moreover, this reality is also compatible with comparative studies in the field, which confirms that the family has a crucial role as a first caregiver, especially in the domain of mental health [45]. Furthermore, some studies were carried out during the COVID-19 pandemic and empirically confirmed the relationship between family support and resilience, particularly in reducing the effects of COVID-19 on mental health through emotional creativity on the part of those charged with healthcare [7,18–29].

The second result is that the father, although he generally holds the role of providing financial assistance, does not have a predominant position in the life of the child who has a mental illness because there is the involvement of the mother on the one hand, and all the other members of the family on the other hand, which explains and confirms the second hypothesis according to which the child within the family must have support, even within low-income families, because of the emotions that matter in conditions of risk such as those of COVID-19. Indeed, researchers have confirmed emotional support's importance more than informational or tangible support [29,30].

In this context, we want to mark the difference between individual resilience and the resilience of children. They are not the same. Indeed, others do not consider individual resilience. It can be represented individually by the father, the mother, or the child (individual resilience). This resilience cannot lead to the realization of the resilience of the sick child.

On the other hand, the child's resilience (personal resilience) can be achieved through the involvement of family members, which confirms the relationship between the child's resilience and the social capital [46–48], simultaneously with the strength of the mother and the father (as roles, not as individuals). It can extend even to other family members and neighbors. Furthermore, individual resilience has no family reference (no role), even if it is practiced within the family. It cannot have this connectivity with other family members.

## 5. Conclusions

This study highlighted the importance of social institutions such as the family; in particular, if there are factors which multiply the risks; for example, the mental illness of a member of the family, that the member is a child, and that their age does not exceed 16. The third and most influential risk is that of the total environment represented by COVID-19. These circumstances attract the attention of researchers, especially psychosociologists, who aim to deepen this research by measuring the emotional support these children can have, not only within the family but also in the institutions responsible for providing social and health assistance with the goal of manufacturing the effect of the family for those with no family.

It has been shown with this study that the family formed the cornerstone in providing social support and achieving the protection of individuals and families through resilience via implicit social relations that constitute the fundamental guarantee of the continuity of these relationships. These results require careful future research to clarify the function of social capital in increasing trust and reciprocity within the family institution, which would prepare children for the future based on the care and adjustment made for them.

## 6. Recommendations

The recommendations that we can make to the actors responsible for social policies are to take care of the family institution in order to make it one of the positive factors in the life of the child and, at the same time, reduce violence, poverty, and social and psychological damage that may occur in other families and turn the institution into a negative factor. In

addition, families may revive the social relations linked to the social capital, such as their ties with extended family, friends, and neighbors, to benefit from social and psychological assistance which may have a positive impact on children suffering from mental illnesses, especially in times of crisis.

**Funding:** This research received no external funding.

**Institutional Review Board Statement:** I conducted the study following the academic code of ethics that is being worked on in the university in which I work in Morocco, such as Informing the respondents about the subject of the study, obtaining their consent to participate in the study, and guaranteeing their right to withdraw whenever they want, in addition to undertaking to ensure the confidentiality of the information stated by the respondent, and not exposing him to any material or moral harm related to his participation in the research.

**Informed Consent Statement:** Not applicable.

**Data Availability Statement:** Data in the archives were not resorted to, but rather a sample of the respondents was resorted to, and the obtained field information was unloaded. Its results are calculated, and the results and data attained are included in the study publicly.

**Conflicts of Interest:** The author declares no conflict of interest.

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
