# Peer review of "Family Social Support and Children’s Mental Health Resilience during COVID-19—Case of Morocco"

_2673-995X, doi:10.3390/youth3020037_

Round 1

Reviewer 1 Report

Please read the attachment. Thank you.

Author Response

Thank you for reading my article and making comments that improve it. Please see the attachment.

Reviewer 2 Report

This is an interesting read, especially the resilience sections exploring the array of literature and types explained. The applied research design with included variables offered validity to the reasoning of the end results. The suggested hypotheses enabled focus to the possible outcomes of the findings.

The sections were comprehensive and widely sourced from an array of literature offering the complexity of the subject matter (resilience). The findings highlighted the significance of roles within support structures including the differing types. The findings presented some correlation with the original hypotheses but with added depth.

Observations:

- the article while comprehensively informative could potentially form two articles as each main section was lengthy. Could the resilience section be reduced without losing its significance and relevance?

- there were some slight grammatical issues such as the use of 'tense'. Some parts presented in future tense with others in past tense. Suggest all is in 'past' tense.  Another proof read may offer a review of the writing execution.  

Author Response

(The authors gave the same response as above.)

Round 2

Reviewer 1 Report

The authors have answered my questions.

Thank you.